# An Appraisal of Potential for Sowing of *Nasturtium officinale* into Streams to Mitigate Nutrient Pollution in Eastern Scotland

**DOI:** 10.3390/ijerph17030895

**Published:** 2020-01-31

**Authors:** Andy Vinten, Patrick Bowden-Smith

**Affiliations:** 1James Hutton Institute, Craigiebuckler, Aberdeen AB15 8QH, Scotland, UK; 2Pittarthie Farm, Anstruther, Fife KY10 2RZ, Scotland, UK; cbowdensmith@aol.com

**Keywords:** watercress, stream nutrient retention, diffuse pollution mitigation

## Abstract

This study examines a farmer-led initiative to sow watercress (*Nasturtium officinale*) in field ditches. The objective was to assess the potential of this practice to mitigate summer nutrient loads in rivers. Two ditches—one seeded, the other unseeded—on a mixed-livestock farm in Eastern Scotland were monitored during the spring-summer of 2014–2016. The un-replicated trial design limited statistical analysis. However, changes in N and P concentrations along the two ditches were measured. In the watercress-seeded ditch, N retention of 0.092 g/m^2^/d (*p* < 0.001, SE = 0.020) and P retention of 0.0092 g/m^2^/d (*p* = 0.001, SE = 0.0028) occurred, while total organic C in the water increased along the ditch. Retention was close to zero for the unseeded ditch. The seeded ditch was also found to have more dry matter production and lower stream temperature. The impact of plastic covering (to increase spring temperature) on vegetation and nutrient removal was also assessed on replicate 5-m sections of the ditches. No significant impact on N and P removal was found; however, the release of C increased significantly in the plastic-covered sections. The rise in air temperature (up to > 30 °C) promoted a greater growth of opportunist species (nettle (*Urtica*), rush (*Juncus*), and grasses. These observations were used to make a simple assessment of the potential catchment scale impact of seeding watercress into first and second order streams in the nearby Lunan Water catchment. It was concluded that this could make a significant contribution to the reduction of nutrient loads.

## 1. Introduction

It has been well established that macrophyte-associated nutrient retention in watercourses is an important sink for diffuse pollutants such as nitrogen (N) and phosphorus (P). Harvesting of macrophytes also has the potential for consumption or recycling to land [1,2]. Macrophytes regulate stream function via direct uptake of nutrients from water, by providing a substrate for epiphytic biofilms [3] and by slowing flow in both the water column and the hyporheic zone [4]. However, in some studies, workers have reached the conclusion that macrophytes have a large impact on trophic state in streams, but offer little potential to influence nutrient removal via management [5].

Watercress (*Nasturtium officinale*) is tolerant of a wide range of nutrient concentrations found in freshwater streams and is often a dominant member in streams with high nutrient loads [6]. Experiences in New Zealand [7,8,9] suggest that the luxury uptake of N by watercress (up to 4% of dry matter) may lead to substantial removal of N from polluted drainage water, particularly if it is harvested regularly. Field experiments have shown that young watercress plants are able to absorb N at a higher mass-specific rate, as compared to older plants [7]. 

In the interest of promoting citizen science [10], it is valuable to make use of farmer-led initiatives for pollutant mitigation, recognising that proactive uptake of government-led initiatives can be poor [11]. There is also the potential to be monetarily compensated on showing the required results [12] in an effort to boost the delivery of ecosystem services. A farmer-led initiative to sow watercress (*Nasturtium officinale*) in field ditches was appraised in this study. The objective was to assess the potential of seeding to mitigate summer N and P loads in river water. Two ditches—one seeded, the other unseeded—on a mixed-livestock farm in Eastern Scotland were monitored during the spring-summer of 2014–2016. The two ditches were on either side of a single grassland field in an upland, but relatively dry part of Eastern Scotland, at Pittarthie, an organic farm owned and managed by an innovative farmer with strong interest in ecological management [13]. The farmer also had prior experience and interest in growing watercress in polytunnels. The effect of enhancing the temperature of water to promote growth in spring/early summer using temporary plastic coverings over the ditches was also considered. Assessment of reduction in N and P loads along the ditches for six sampling dates over 2014–2016, and measurements of plant dry matter and N and P uptake were carried out, although the lack of replication of ditch treatments limited the statistical rigor of inferences from observations, highlighting an important potential limitation of such citizen science.

Diffuse pollution is a significant problem; it led to the failure of the water environment in Scotland achieving Good Ecological Status, as required by the EU Water Framework Directive [14]. Evidence is needed of the comparative effectiveness of potential mitigation measures, and Diffuse Pollution Monitored Catchments (DPMCs) were therefore established in Scotland to assess such measures at a catchment scale. One of these DPMCs, established in 2006, is the Lunan Water, a 134-km^2^ catchment in Angus, Eastern Scotland. It is a typical lowland, mixed arable farmland catchment. Water flow and quality data has been collected in this catchment for >10 years [15]. A simple framework to assess the effectiveness and cost-effectiveness of a range of potential mitigation measures for P runoff from land ([16,17]), using a “smart” export coefficient approach, has also been developed. However, this approach did not consider retention of nutrients by aquatic vegetation. It was thus decided that it would be of great value to include this in the cost-effectiveness framework prescribed in this paper. Based on observations at Pittarthie, this study made a preliminary assessment of the potential impact of watercress seeding in the Lunan Water catchment. 

## 2. Materials and Methods

### 2.1. Study Area for Watercress Seeding Trial

Pittarthie is a small, mixed livestock, 170-ha family farm located in eastern Fife, Scotland. Conservation of the water environment is a key theme at the farm, surrounding ponds, and wetlands. Fish ladders and silt traps were thus installed and other measures taken to control erosion via water margins. The monitoring reported here concerns two open ditches, which run down opposite edges of a south facing grass field that was cut for silage/hay production. One of these ditches was previously seeded with watercress. The ditches have a nominal 0.5-m wetted width during summer and drain into a small stream that runs into the Kinaldy Burn (has a Water Framework Directive ecological status of moderate). The location of these ditches is shown in Figure 1. 

Ditch 1: Has natural growth of uncontrolled aquatic vegetation. Estimated catchment area of the ditch = 5.2 ha. Length of the ditch below the highest sampling point = 280 m. Mean slope = 0.033 m/m.

Ditch 2: Was seeded with watercress in 2013. Estimated catchment area of the ditch = 2.6 ha. Length of ditch below the highest sampling point = 280 m. Mean slope = 0.056 m/m.

In the spring of 2015 and 2016, a series of five 5-m longitudinal segments along the length of each ditch were covered with plastic polytunnel sheet with the objective of increasing spring air temperature and enhancing vegetative growth (Figure 2). The plastic was laid over a wire frame mounted around fence posts, weighed down with wooden battens along the sides of the ditch. These lengths were separated by uncovered sections. 

### 2.2. Estimation of Nutrient Retention by Ditches

Water was sampled in transects along the ditches on three dates in 2014, one date in 2015, and two in 2016. Details of the water sampling are shown in Table 1. The samples were analysed by standard methods [15] for pH, EC, NH_4_-N, NO_3_-N, Total N, Org N, PO_4_-P, Tot-P, Org-P, Total Organic C, Alkalinity, K, and Cl. 

The daily nutrient loading L_N,P_ (g/d) at any sampling point distance z (m) along the ditch was estimated using Equation (1):L_N,P_(z) = 86400Y_N,P_ (z) Q_ditch_(1)
where: Y_N,P_(z) = N,P concentration in sample (mg/L); Q_ditch_ = mean daily flow in ditch (m^3^/s).

The value of Q_ditch_ was estimated for each sampling date by scaling the observed daily flow data for the River Eden at Strathmiglo (56°16′39.4″ N 3°15′11.2″ W) (obtained from the National River Flow archive [18] by the ratio of the area of the Eden catchment to the estimated area of the ditch catchment [19]: Q_ditch_ = Q_Eden_ A_ditch catchment_/A _Eden catchment_(2)
where: Q_Eden_ = mean daily flow in River Eden (m^3^/s); A_ditch catchment_ = catchment area of ditch estimated from field contours (m^2^); A_Eden catchment_ = catchment area of River Eden as stated in the National River Flow Archive (m^2^).

The impact of ditch retention on nutrient loading at each sampling time t, S_N,P_(t) (g/m^2^ ditch/d), was estimated from the slope of the regression line of ΔL_N,P_ = L_N,P_ (z) − L_N,P_(0) against distance z (m) down the ditch, and the assumed width of the wetted area of ditch, W (m):S_N,P_(t) = Δ L_N,P_ /WΔz(3)

The changes in NO_3_-N, PO_4_-P (SRP), and total dissolved organic C (TOC) loads from top to bottom of the ditch were analysed with GENSTAT (19^th^ edition) using general linear regression for groups. A students-t test was used to compare the slopes of these lines. Inspection of the data showed that some water samples had anomalously high K and/or Cl values, which we took to be evidence of the direct impact of fertilisers or manure. These sample data were removed from statistical analysis. 

The impact of covering with plastic on water chemistry was analysed using ANOVA by comparing the change in solute concentration between successive sampling positions (UU: uncovered to uncovered; UC: uncovered to covered; CC: covered to covered; CU: covered to uncovered). 

### 2.3. Estimation of Plant Nutrient Uptake

Nutrient retention is only partially controlled by plant nutrient uptake. In order to estimate plant nutrient uptake and compare it with the overall nutrient retention in the ditches, plant biomass (leaves and stems combined) was sampled in June 2015 and May 2016 from five 3-m sections of the ditches at successive sampling positions in the covered (C) and uncovered (U) parts. Width of the vegetation cover across the nominal 0.5-m wetted width of ditch was noted. Above-ground dry matter content of watercress and other vegetation was determined by weighing samples before and after drying in an oven at 60 °C. The mean and standard deviation of dry matter for the covered and uncovered sections of the ditches were determined. For 2015 only, sub-samples of the vegetation samples were analysed for total N using a Thermo Finnigan Elemental Analyser, Thermo Fisher Scientific, Hemel Hempstead, UK (FlashEA 1112 Series).

Total P determination was done using inductively coupled plasma optical emission spectrometry (ICP-OES) analysis of a nitric acid digest of plant material. Total Plant uptake of N and P from seeding to the sampling date was calculated using Equation (4): UPT_N,P_ = 100 × X_N,P_ × DM% × FW(4)
where: UPT_N,P_ = N,P uptake (mg/m^2^ stream surface); X_N,P_ = %N,P in dry matter of plant material (-); DM% = %DM in samples of fresh plant material (-); FW = weight of fresh plant material harvested per m^2^ of stream surface area (kg/m^2^).

Note that for 2016, mean X_N,P_ data from 2015 were used to estimate uptake, as there are no measurements of nutrient content of plant material. In addition, as the ditches are unreplicated, it is not possible to draw firm statistical inference from dry matter and nutrient uptake data, and no statistical test has been conducted on this data.

Temperature of the air and ditch water in each covered and uncovered section was noted on the sampling dates. In June 2016, the percent cover associated with watercress, grasses, other vegetation, and open water was determined by a quadrat on five 3 m lengths for the covered and uncovered sections in each ditch. 

### 2.4. Appraisal of Potential for Watercress Seeding in the Lunan Water Catchment

To appraise the potential of watercress seeding, retention rate data from the Pittarthie trials were extrapolated to a catchment scale study in another mixed farming area in Eastern Scotland, the Lunan Water. Intensive monitoring of stream chemistry and flows has taken place for >10 years [15]. This catchment is of interest because within the upper catchment are two Lochs—Rescobie and Balgavies, covering 1.78 km^2^—which suffer from over-enrichment of nutrients, leading to serious eutrophication in summer and also affecting the Lunan Water downstream [20]. In addition, much of the catchment is underlain by porous groundwater bodies, vulnerable to nitrate pollution, which then reconnects with surface waters further downstream. Rescobie Loch had annual geomean values for TP (2003–2006) of 70.1 µg/L, which implies a mean TP loading of 0.27 kg/ha catchment/year. The target reduction in TP loading to achieve Good Ecological Status is 0.17 kg P/ha catchment/year. 

The loading of P and N (load_P,N_) to the Lochs from the monitored Lunan sub-catchments was estimated using a combination of discharge estimation from 15 min water level data, 15 min turbidity monitoring, storm event sampling, and fortnightly spot samples [21]. For the present work, estimates of concentrations and loads were made for April 2010 to March 2012 for three sub-catchments: Lemno Burn, Balgavies Burn, and Baldardo Burn (Figure 3). Loads of TP were estimated using 15 min turbidity data and storm event-based calibrations of TP vs. turbidity [15,22]. Loads of NO_3_-N and Soluble Reactive Phosphorus (SRP) were estimated by interpolating fortnightly spot sampling and discharge data using the unbiased Beale estimator [15]. 

The impact of stream nutrient retention on nutrient loads to the loch, Δload_N,P_, was estimated using the following equation: Δload_N,P_ = S _N,P_(t)Δt A_stream_/A_catch_(5)
where: Δload_N,P_ = change in N,P load from catchment to loch resulting from watercress-mediated retention (g/m^2^ catchment) over a time period; S _N,P_(t) = retention rate of N,P by watercress (g/m^2^ stream surface/d) (estimated from Equation (3)); A_stream_ = stream surface area in catchment (m^2^ stream); A_catch_ = area of catchment (m^2^ catchment); Δt = time period(d). 

## 3. Results

### 3.1. Nutrient Retention

For both NO_3_-N and SRP, water quality was well within the acceptable standards for rivers (Table 2). Nonetheless, these headwater concentrations contribute to nutrient loads lower down the catchment, where concentrations are higher.

Decline of SRP loads compared with estimated input loads along the seeded and unseeded ditches is shown in Figure 4a. All data are shown, except for June 2015 when there were large unexplained fluctuations in P load down the ditch, and between the first and second sampling points down the ditch for the seeded plot in August 2014 and the unseeded plot in May 2014. The high P concentrations in some of these samples correlated with high K concentrations, suggesting a direct fertiliser impact on the concentrations observed. There may have been issues with entrainment of sediment when sampling at low flows, which could affect P concentrations. On omitting these data from the analysis, general linear regression with groups shows a significant (*p* = 0.005) impact of seeding vs. non-seeding P removal rate (seeding: −0.0092 g P/m^2^ ditch/d, *p* = 0.001; no seeding: 0.0024 g P/m^2^ ditch/d, *p* = NS). 

Load reduction of NO_3_-N is shown in Figure 4b. There was an anomalously large reduction in N load between the first and second sampling points down both ditches for May 2014 and also for the unseeded ditch in March 2014. Omitting these data from the analysis, general regression analysis with groups shows a significant (*p* = 0.004) impact of seeding vs. non-seeding on N removal rate (seeding: −0.092 g/m^2^ ditch/d, P < 0.001, SE = 0.010; no seeding: −0.022 g N/m^2^ ditch/d, P = NS). There were also several examples of a significant upward trend in TOC loads in both the seeded and unseeded ditches (see Appendix A).

### 3.2. Effect of Plastic Covering

The temperatures observed at the time of sampling in June 2015 and May 2016 are summarised in Table 3. The plastic covering greatly increased the daytime air temperature, particularly in the seeded ditch in June 2015. Water temperatures were lower on both dates in the seeded ditches than in the unseeded ones. Covering did not have a significant impact on water temperature. 

The impact of plastic covering on water chemistry was analysed using ANOVA by comparing the change in solute concentration between successive sampling positions (UU: uncovered to uncovered; UC: uncovered to covered; CC: covered to covered; CU: covered to uncovered). No significant impact of the different transitions on change in water chemistry was found, except for a higher release rate of TOC in the covered sections.

### 3.3. Growth and Nutrient Uptake

Shoot dry matter yield in June 2015 and May 2016 is shown in Figure 5. There was a large difference between seeded and unseeded ditches in 2015 and although most of the dry matter in the unseeded ditch was associated with watercress, there was significantly more watercress growth in the seeded ditch. There was slightly more growth of watercress in the covered than in the uncovered ditches. In 2016, there was less growth of watercress in the covered sections than in the uncovered sections of the seeded ditch. The N and P content of the above-ground plant material in 2015 is shown in Figure 6. No distinction was made between watercress and other vegetation in the analysis. Nitrogen uptake was around 25 kg N/ha of wetted ditch for the seeded ditch and P uptake was around 2 kg P/ha. 

The % cover of watercress, grass species, and other covers in the ditches in May 2016 is shown in Figure 7. This also shows that there was much less watercress growth (average 19% cover) in the covered than the uncovered (33% cover) seeded ditches. The covered sections of the unseeded ditch had similar watercress cover to the covered seeded sections. The plant species that invaded the ditches was a mix of grasses, dicots, and rushes. Although we did not measure species content in 2014, the seeded ditch was a more uniform stand of watercress in 2014, the year after seeding. Appendix A shows examples of growth in the ditches.

The N uptake rates per unit of shoot dry matter were estimated for the two dates when dry matter data were available. These were 4.2 and 6.4 mg N/g DM/d for June 2015 and May 2016, respectively. The P uptake rates were 0.09 and 0.2 mg P/g DM/d for June 2015 and May 2016, respectively.

### 3.4. Potential Impact of Watercress Seeding in Upper Lunan Water Catchment

To make a preliminary assessment of the benefits of nutrient retention by seeding of watercress into streams and ditches in the Lunan Water catchment, the values of S _N,P_(t) were assumed to be 0.09 g N/m^2^/d and 0.009 g P/m^2^/d for 4 months (April to July) and 0 during the rest of the year. These broad assumptions could be improved with a better understanding of seasonality of retention and growth of watercress, and by considering the impact of stream velocity on retention [22]; however, they should illustrate the potential of watercress seeding. By assuming a mean water surface area to catchment area ratio of 0.004 (using primary and secondary stream data from Figure 3), the impact of watercress seeding on N and P retention can then be estimated. Watercress requires suitable, low slope water courses, which maintain significant flows, and much of the upper Lunan Water catchment is of low slope, and hence amenable to macrophyte growth (Table 4). Note that the potential P retention exceeded the P load in the Balgavies sub-catchment during spring. If these values are upscaled to the Rescobie Loch catchment (Acatch = 1960 ha), the estimates of load reduction by watercress seeding across the primary and secondary streams are found to be 85 kg P (about 26% of the target P load reduction for the loch to achieve Good Ecological Status for P) and 852 kg N.

## 4. Discussion

### 4.1. Comparison with Other Work on Nutrient Retention by Macrophytes

Our results are in keeping with the effect of macrophytes on nutrient retention observed by others. For example, Vincent and Downes [3] found a 6–11% loss of N mass over a 100-m reach with watercress present, whereas no loss of N mass was measured over the same reach cleared of watercress. In the same Whangamata stream in New Zealand, estimated summer potential N and P retention rates were 0.41 mg N/m^2^ watercourse/day and 0.019 mg P/m^2^ watercourse/day, respectively [8] (assuming 100% cover, area of river reach = 6000 m^2^ and mean flow of 70 L/s). Transient storage was approximately four times larger with watercress present compared with the same reach without the plants. N uptake rates by roots [6] were equivalent to 1.43 mg N/g shoot DM/d in summer (water temperature = 13 °C) and 0.52 mg N/g shoot DM/d in winter (water temperature = 10 °C). P uptake rate was 0.20 mg P/g shoot DM/d in summer. The uptake rate from the stream also depended on the mean percent cover of watercress, which varied from 20% (winter) to 42% (summer). In a range of natural streams in Western Europe [22], retention for N at low stream slopes was 0.3–0.5 g/m^2^/d and for P it was 0.01–0.02 P/m^2^/d; however, with higher slopes (and therefore shorter residence times), the retention rates were smaller. Seeding of streams is therefore likely to be most suitable where ditch slopes have shallow gradients. However, check-dams can be used to manage steeper sites and offer the advantage of promoting settling of sediment in pockets along the ditch length. Danish integrated buffer zones (IBZs) have shown NO_3_-N removal rates of 0.3-0.5 g N/m^2^/d with efficiencies decreasing from 80% to 20% as N loads increased from 1 to 4 g N/m^2^/day, but very variable and even negative P removal efficiencies occur [6].

### 4.2. Factors Affecting Nutrient Retention

It is important to note that only part of the impact of vegetation on nutrient retention is associated with plant uptake. For example, the average P uptake by the shoot material harvested in June 2015 was 0.18 g P /m^2^ in the seeded ditches and 0.05 g P/m^2^ for the unseeded ditches, equivalent to only 20 d of the observed nutrient retention rate. Accounting for root growth may increase the uptake figures by about 30% [7], but it is well known that the presence of vegetation also enhances retention of particulate material flowing past (including organic particulate material from periphyton), which may have a larger impact than the plant uptake itself. For example, a pattern of negative retention of P in summer was observed in the absence of submerged macrophytes but an average of 15% retention over April-August in the presence of macrophytes [23]. The average N uptake by the shoot material harvested in June 2015 was 2.2 g N/m^2^ in the seeded ditches and 0.64 g N/m^2^ for the unseeded ditches, equivalent to 24 d of the observed nutrient retention effect. Dissimilatory N reduction may be enhanced by the presence of C released from plants [24]. The observed release of C may also help to promote immobilisation of N and P in periphytic microbial biofilms [25]. It should be noted that watercress growth may extend into the non-wetted cross section. Here, it may not directly remove water from the stream, but from soil water, and so increase the amount of nutrient uptake [23]. 

It is also important to consider the annual pattern of N and P retention by the watercress. For example, in the Whangamata stream in New Zealand, there was an evident peak in uptake of N and P in the summer months [7]. This stream was spring fed and temperature of the water only varied from 10˚C in winter to 13˚C in summer. This suggests that the annual pattern of uptake may be more related to growth stage and light than temperature. N and P retention in the Pittarthie ditches occurred earlier in the year and the highest N retention values were larger than that of the Whangamata stream, for similar levels of plant cover. This probably reflects the higher N concentrations in the water in the Pittarthie ditches and possibly the longer day length in late spring [26].

### 4.3. Other Management Considerations

Several other factors need to be considered when adopting the practice of watercress seeding in ditches and streams for water quality enhancement. For example, natural mustard oils present in watercress grown on farms may be released during disturbance (e.g., harvesting) and have a deterrent effect on invertebrates [27] such as *Gammarus* shrimps (Worgan and Tyrell, 2005, cited in [28]). However, for undisturbed beds, it is highly unlikely that the effects would be significant [29]. 

The growth of macrophytes also interacts strongly with a hydrological regime. Macrophyte cover is lowest in streams with high flow variability and highest in streams with long duration of low flow and low flow variability [30]. There is also evidence that increased growth of macrophytes will enhance water levels upstream [22]. If macrophytes are not removed during the high flow periods in the upper reaches of this catchment, this may promote retention of water during flood periods. This should be a benefit to downstream riparian owners, who suffer from periodic flooding, but it may not be beneficial to those in the upper catchment, who would appreciate better drainage of flood waters [31]. For this reason, seeding was restricted to first and second order streams, where unwanted flooding impacts would be less likely, although farmers would need to come to terms with less-efficient field drainage. 

Farmers’ perceptions of the importance of stewardship of water quality are increasing globally [32,33,34]. This is important for the achievement of both local enhancement of water quality (farm scale) and for downstream water users, and was an important driver of the research described here. Generating quantitative information has helped to support pro-environmental behaviour. It also helps to ground aspirations to improve water quality in the science challenge of demonstrating benefit. However, the un-replicated design of two ditches—one seeded and the other unseeded—limited the statistical rigour, and highlights the value of co-construction of trials between land managers and scientists, prior to implementation. 

## 5. Conclusions

Watercress planting in drainage ditches on grassland under Scottish lowland conditions enhanced the retention of N and P during the growing period between April and August, compared with natural ditch vegetation (by 0.092 g N/m^2^ wetted ditch/d and 0.0092 g P/m^2^ wetted ditch/d). Only part of this retention can be explained by plant nutrient removal. Plastic covering of ditch sections had a negative impact on watercress growth. The high air temperature (up to >30 °C) promoted greater colonisation by opportunist species such as nettles (*Urtica*), rush (*Juncus*), and grasses. This way of enhancing spring aquatic plant growth is not recommended.Estimates of potential watercress-mediated N and P retention for the Lunan Water catchment showed that loading into the eutrophic Rescobie and Balgavies Lochs in the upper catchment could be reduced by about 26% of the required load reductions, by targeting first and second order streams for watercress seeding. Seeding of streams is most suitable for shallow slopes with steady water flow. Other factors need to be considered before adopting the practice of watercress seeding in ditches and streams for water quality enhancement, including potential re-release in periods of high flow, impact on stream ecology.Ecology of release of natural mustard oils, impact on upstream water levels, and cost-effectiveness compared with other mitigation methods. The farmer-led approach to use watercress-seeded ditches to mitigate diffuse pollution, supported by science-based evidence, has the potential to facilitate its adoption on a wider scale.

## Figures and Tables

**Figure 1 ijerph-17-00895-f001:**
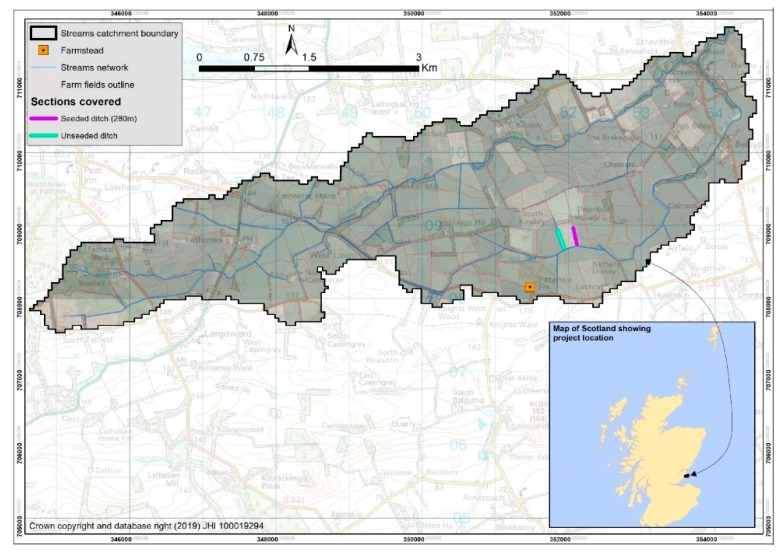
Location of the Pittarthie Farm watercress trial. Grid Reference No. 51505 08136.

**Figure 2 ijerph-17-00895-f002:**
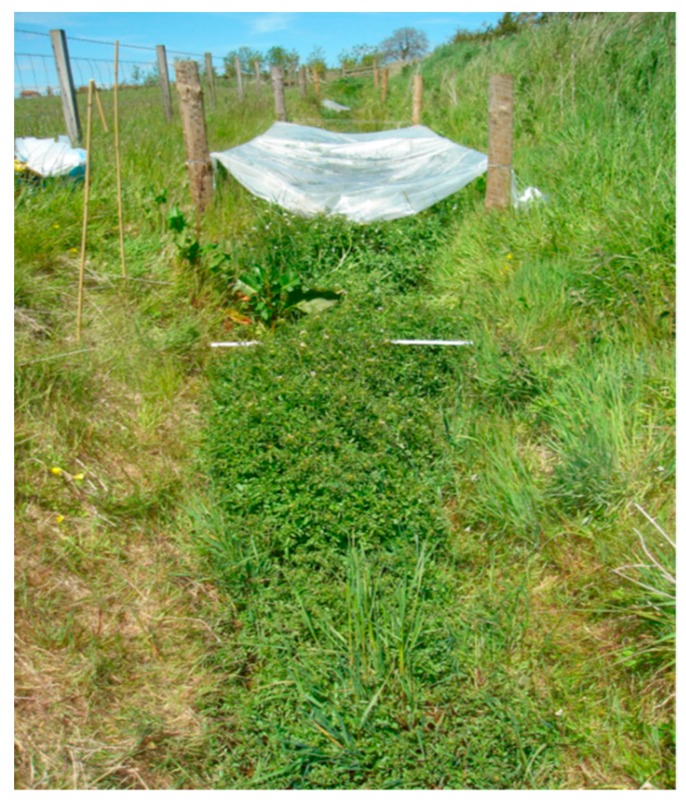
Seeded ditch in May 2014, showing uncovered and plastic-covered segments of the ditch.

**Figure 3 ijerph-17-00895-f003:**
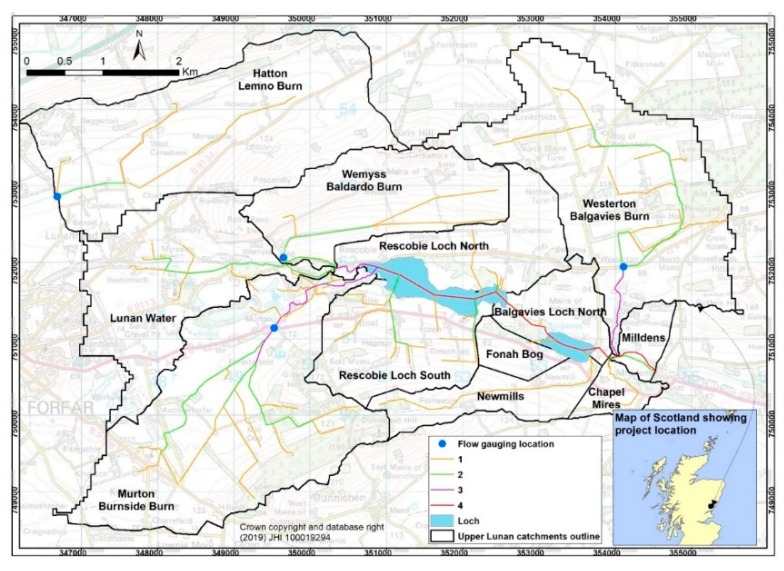
Sub-catchments in the upper Lunan Water. The flow gauging station/watercourse for the three monitored sub-catchments are: Westerton/Balgavies Burn (A_catch_ = 590 ha), Wemyss/Baldardo Burn (A_catch_ = 238 ha), and Hatton/Lemno Burn (A_catch_ = 710 ha). The total catchment area of Rescobie Loch is 2016 ha. The line colouring shows the stream order. Regular water chemistry sampling takes place at the three flow gauging stations and the SEPA run station at Murton/Burnside Burn.

**Figure 4 ijerph-17-00895-f004:**
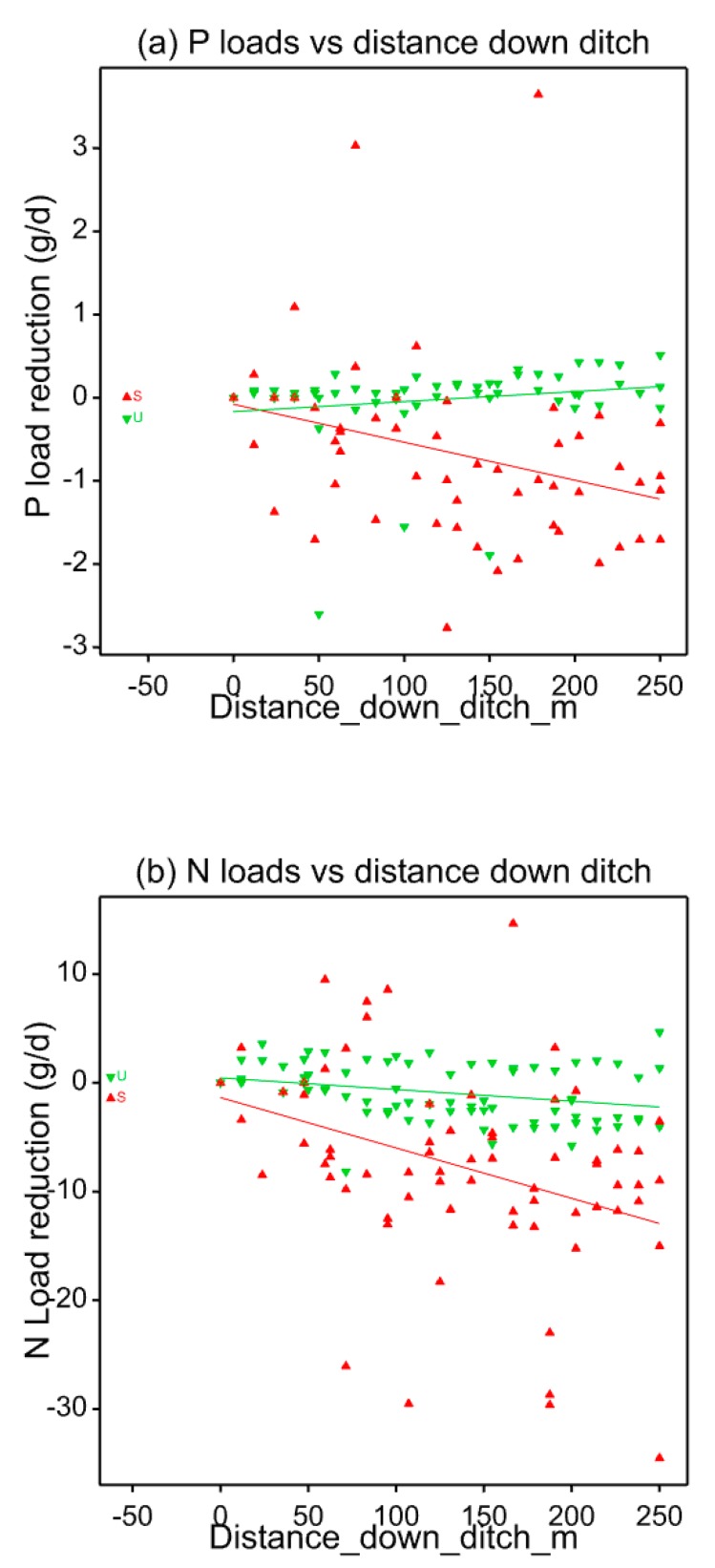
Estimation of (**a**) SRP and (**b**) NO_3_-N retention rates for ditches seeded with watercress, and those not seeded. Slopes of load reduction for watercress ditches (red) are equivalent to −0.0092 g P/m^2^ ditch/d and −0.092 g N/m^2^ ditch/d, assuming an average ditch width of 0.5 m.

**Figure 5 ijerph-17-00895-f005:**
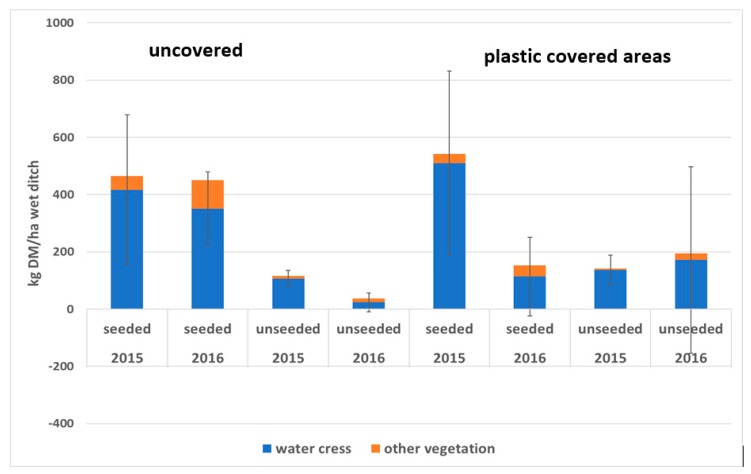
Average dry matter content and SD of above-ground growth in the seeded and unseeded ditches, for June 2015 and May 2016, uncovered and plastic covered areas, per unit wetted area of ditch. Width of the wet ditches is assumed to be 0.5 m. The SD of watercress DM is also shown (*n* = 5).

**Figure 6 ijerph-17-00895-f006:**
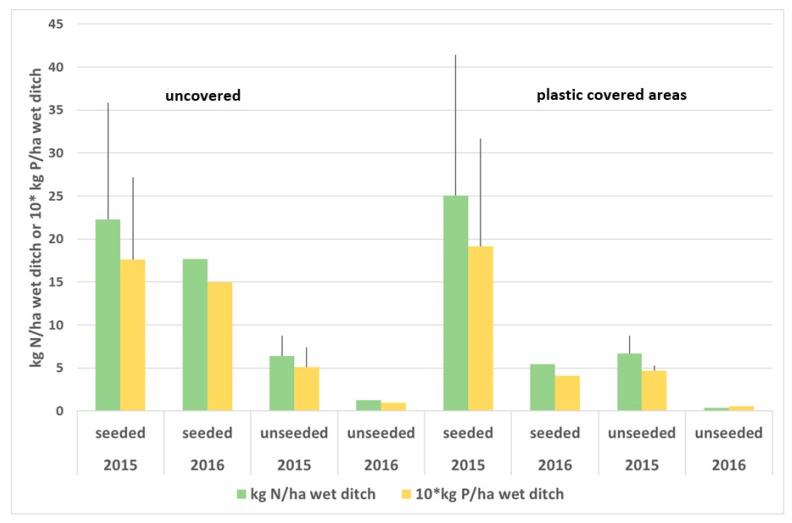
N and P content and SD of shoot of watercress in seeded and unseeded ditches, for June 2015 and May 2016, uncovered and plastic covered areas, per unit wetted area of ditch. Width of the wet ditches is assumed to be 0.5 m. SD is also shown for 2015 (*n* = 5). Note that the 2016 results use treatment mean N and P content from 2015 to estimate uptake.

**Figure 7 ijerph-17-00895-f007:**
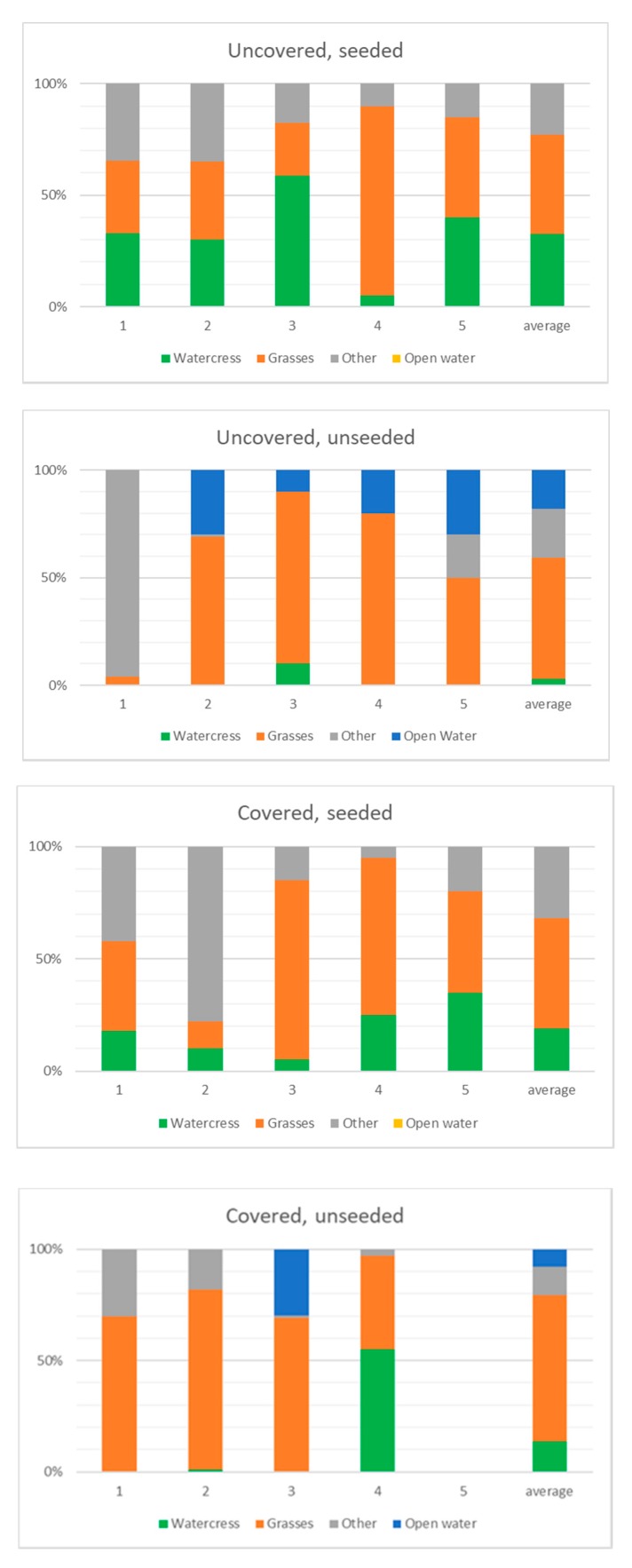
Percent cover of watercress, grasses, other plants, and open water in covered and uncovered, seeded and unseeded sections of ditches in May 2016. Other plants are mainly nettle (Urtica dioica), rush (Juncus effusus), with some Rumex, Violaceae, Galium, Ranunculus, Equisetum, Epilobium, and Onopordon spp.

**Table 1 ijerph-17-00895-t001:** Timeline of plastic-covered treatments and sampling of ditches at Pittarthie Farm, 2014–2016.

Year/Month		Water Samples	No. of Samples along Seeded, Unseeded Ditches	Plastic in Covered Sections	Vegetation Samples
2014	March	13/03/2014	5,6		
2014	May	30/05/14	5,6	Covered on 26/5/14	
2014	August	14/08/14	5,6		
2015	June	10/06/15	18,20	Covers removed after sampling	10/06/2015
2016	April	28/04/16	22,22	Covered on 28/4/16	
2016	May	30/05/16	20,20	Covers removed after sampling	30/05/2016

**Table 2 ijerph-17-00895-t002:** Mean and standard deviation of NO_3_-N and SRP concentrations in ditch and receiving stream samples.

SRP (μg/L)	2014		2015		2016	
	Mean	SD	Mean	SD	Mean	SD
Seeded ditch	42	52	30	30	52	26
Unseeded ditch	49	38	27	19	33	6
Receiving stream	28	10			15	2
NO_3_-N(mg/L)						
Seeded ditch	1.6	0.7	1.5	0.3	4.7	0.2
Unseeded ditch	2.0	1.2	0.6	0.1	1.5	0.3
Receiving stream	1.0	0.6			1.7	0.6

SD = standard deviation.

**Table 3 ijerph-17-00895-t003:** Mean air and water temperatures for the covered/uncovered sections of the seeded/unseeded ditches. Spot temperatures at time of sampling (*n* = 4 or 5).

			June 2015		May 2016	
			Air	Water	Air	Water
SC	Seeded	Covered	31.7	11.4	16.0	9.1
SU	Seeded	Uncovered	22.5	11.2	14.7	8.9
UC	Unseeded	Covered	29.0	16.4	17.0	11.1
UU	Unseeded	Uncovered	n/a	17.1	16.4	11.0

**Table 4 ijerph-17-00895-t004:** Estimated impact of watercress seeding on P and N loads in upper Lunan Water sub-catchments.

Sub- Catchment	Season	TP Load kg/ha		% Reduction in P Loads by Seeding		NO_3_-N Load Kg/ha		% Reduction in N Loads	
		2010–2011	2011–2012	2010–2011	2011–2012	2010–2011	2011–2012	2010–2011	2011–2012
Lemno	autumn	0.20	0.25	0.0%	0.0%	7.7	9.7	0.0%	0.0%
	winter	0.59	0.13	0.0%	0.0%	17.6	9.6	0.0%	0.0%
	spring	0.12	0.40	26.1%	7.9%	7.3	4.9	4.4%	6.5%
	summer	1.32	0.20	0.9%	5.8%	24.8	11.3	0.5%	1.0%
	annual	2.24	0.99	1.9%	4.4%	57.4	35.5	0.8%	1.2%
Baldardo	autumn	0.19	0.08	0.0%	0.0%	8.7	6.9	0.0%	0.0%
	winter	0.27	0.06	0.0%	0.0%	17.8	6.5	0.0%	0.0%
	spring	0.04	0.12	83.0%	25.9%	4.5	4.5	7.2%	7.1%
	summer	0.11	0.07	10.4%	17.3%	9.2	8.5	1.2%	1.4%
	annual	0.60	0.33	7.3%	13.1%	40.1	26.4	1.1%	1.7%
Balgavies	autumn	0.12	0.04	0.0%	0.0%	7.3	4.3	0.0%	0.0%
	winter	0.18	0.02	0.0%	0.0%	23.6	5.3	0.0%	0.0%
	spring	0.01	0.03	298.4%	93.9%	4.5	2.8	7.1%	11.4%
	summer	0.07	0.03	15.7%	34.5%	7.8	10.3	1.5%	1.1%
	annual	0.39	0.13	11.3%	34.8%	43.2	22.7	1.0%	1.9%

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
