# Peer review of "An Appraisal of Potential for Sowing of Nasturtium officinale into Streams to Mitigate Nutrient Pollution in Eastern Scotland"

_ijerph, 2020, doi:10.3390/ijerph17030895_

Round 1

Reviewer 1 Report

Authors conducted the revisions adequately. The paper can be accepted for the publication. 

Author Response

no response

Reviewer 2 Report

After reading the Letter of response and the new version of the manuscript I think that the manuscript has been improved (mostly the discussion section). However, I saw that some of my comments have not been addressed, thus there is still some work to do.  For instance, I had suggested to clarify the objective(s) of the manuscript which is a must in a scientific paper and authors just removed this part. In some other parts, I had suggested to explain how they measured/calculated some variables such us plant biomass or plant nutrient uptake and these information is still missing in the manuscript. I have also suggested to put some effort on organizing and clarifying the M&M section because I thought that the previous version was confusing. Despite they run some slight changes, I do not see a substantial improvement. In its current version it’s still very difficult to follow the experiments done, it’s rationale and the relationship between them. For instance, I understand the experiment associated to the 2 ditches (unvegetated vs. vegetated or covered vs. uncovered comparisons) but I do not understand the experiment related to the river Eden (lines 98-101). In lines 217-220 there are plant uptake estimations. How did you calculate those values? As I mentioned before, you should explain in the M&M how did you calculate those data. Regarding to the calculations in Lunan Water catchment, I do not understand how did you extrapolated nitrate and SRP concentrations. Could you please explain that calculation in the text? From lines 158 to 220 there are no references of other studies. Since authors include a “Results and Discussion” section, I would expect an explanation of the observed results and a comparison with other studies. However, I can found those comparisons in lines 257 to 275. Therefore, consider a) merge those lines in a single section or b) organize “Results” and “Discussion” as separate parts. 

As I mentioned in my first review, this manuscript has some interesting points but authors need to do an important effort to clarify “what”, “why” and most importantly “how” they done their research. They also have to put some effort to better organize and present the information.

Reviewer 3 Report

Dear Authors,

I accept manuscript in its present form.

Regards,

Author Response

No response

Round 2

Reviewer 2 Report

After reviewed the letter of response and the latest version of the manuscript entitled “An appraisal of potential for seeding of Nasturtium officinale into streams for mitigation of nutrient pollution of water in eastern Scotland” I found a clear improvement. The current text organization is much better and the reader can follow the investigations and the calculations done. I’m satisfied with authors responses’ to comments/suggestions and I have no further comments.  

This manuscript is a resubmission of an earlier submission. The following is a list of the peer review reports and author responses from that submission.

Round 1

Reviewer 1 Report

Dear Authors,

The title of the manuscript “What can watercress do for you? An appraisal of potential of seeding Nasturtium officionale into streams for mitigation of diffuse nutrient pollution of water in eastern Scotland is of interest to the International Journal of Environmental Research and Public Health. I find the paper and your idea that it contains interesting but, in my opinion, in its present form, the paper needs improvements. The main aspects to be reviewed are the following.

General and minor comments 

Title - the text is too long; “ An appraisal of potential of seeding Nasturtium officionale into streams for mitigation of diffuse nutrient pollution of water in eastern Scotland”;

1.

Abstract (L:8-19) – The text should be significantly improved, supplemented by clearly pointing out the targeting research gap, the goal of your research and explaining why your investigation is needed. The main aim of the study (hypotheses L:86-94), results, conclusions should be formulated clearly;

2.

Section „Introduction” – The text is too long and should be rewritten (and supplemented) - essence of the matter should be more emphasized; the issue should be clearly described and explained. Please, enter the author's surname in the place of the number, e.g. L:34 [7]; L:40 [5]; L:48 [6]; L:49 [10] etc. (Please, correct accordingly along the manuscript); The text L: 36-65 should be removed and replaced to new section “Results and Discussion”

3.

Section „Methods” should be combined with „Materials” and changed to „Materials and Methods” (e.g. subsections: „Study area” (L:101-122); „Analysis....” (L:126-175)). In reviewer’s opinion this section requires both an adjustment and an elaboration. The text should be re-edited (more briefly), there are too many subsections. The equation (1) in L:168 – Please, add references and explanation, there is no citation, it is not clear and should be explained;

4.

Section „Results” -  In reviewer’s opinion this section requires both an adjustment and an elaboration (more compacty). Section „Results” should be combined with section „Discussion”. This section is too long (L:177-315, without supplements) and has too many subsections. The text, e.g. L: 178-180; L:188-191; L:273-282 should be removed (or elaborated). Furthermore, sentences (e.g. L: 221; 226; 236; 273, etc.) are begining in the words: „Figure” or „Table” – it should be redrafted (style); same as L:215: „..We analysed..” - Please, avoid personal form. This section should be more supplemented with discussion. The similarities and differences between this study and other findings should be summarized and explained (citation). Please, pay attention on significant relationships.  

5.

Section „Discussion” – the text L:335-337 sounds like ..conclusions, same as in L:344-390 – it should be removed from this section (or elaborated); Moreover, please, avoid personal form (e.g. L:332 „We also..”).

6.

In reviewer’s opinion, section “Conclusions” (L:400-406) is the weakest point of the manuscript. As far as section „ Conclusion” is concerned, the reviewer did not find.. conclusion there; instead, this rather sounds like results. Comprehensive and consistent conclusions are missing. A few sentences of the summary of results are necessary.

Remarks regarding Tables and Figures:

- Description of Figure: 2; 3; 5; 7; 8 should be improved;

- Figure 6 and Figure 8– two graphs as one will be better solution (any repetitions should be avoided), same as in Figure 7, Table 3 (“segment; average”; etc.) and Table 4 (autumn (e.g. 1), winter (e.g. 2), etc.).

Bibliography:

The list of publications should be corrected, standardize and adapt to the requirements of IJERPH (MDPI);

Best regards,

Reviewer 2 Report

This paper investigates the effect of watercress on nutrient mitigation in agricultural landscapes. Authors compared nutrient retention capacity of two ditches, one seeded with this aquatic plant and another one unseeded and investigated how increases in temperature may affect plant growth and community diversity. Finally, they also investigated how observed results may be upscaled in a nearby experimental catchment. These are interesting topics for developing ecologically sound strategies to mitigate nutrient enrichment associate to human activities. After reading the manuscript several times, I have to say that a lot of effort have to be done to improve the organization, clarity, presentation and interpretation of the results. The objectives of the study are extremely confusing, methodology used to calculate the results are often missing and some parts of the result section are apparently out of the scope. Finally, some important parts of the discussion are focused on topics that are far away from the a priori objectives of the manuscript. To this point, I would suggest to first put some effort to clarify the objective(s) of the study, explain the methodology used and improve the organization of the Methods section. Afterwards, let’s discus the observed results and avoid referring to other topics that have not been apparently investigated in this study. Now I will try to develop in more detail my major concerns:

Introduction:

Line 28: I would say “variability” instead of “kinetics” here. Uptake kinetics refers to the response of a given community to an acute increase in nutrient concentration (see Eal et al. 2006 or Ribot et al. 2013).  

Line 34: here another recent study investigating how macrophytes may increase water resident time: Nikolakopoulou, M., A. Argerich, J. D. Drummond, E. Gacia, E. Martí, A. Sorolla, and F. Sabater. 2018. Emergent Macrophyte Root Architecture Controls Subsurface Solute Transport. Water Resources Research.

The last paragraph of the Introduction need some clarification. It’s quite difficult to follow the objective(s) of the manuscript by just enumerating the hypotheses. On the other hand, a hypothesis should be followed by a testable prediction that could be used to corroborate (or reject) it. I would suggest to clearly state the objective(s) of the study and then specify how you had addressed them. Then, you could add some expectations/predictions that may be discussed in the Discussion.  

Methods:

The Methods section need a deep reorganization and clarification. For instance, it’s quite weird that the Study site subsection is at the same level that a subsection explaining flow calculation. Maybe you can add a subsection called “Parameter calculations” and include there all calculations done in this study (i.e., flow, % of N and P reduction, etc...). On the other hand, there are some parameters in the Results section that are not explained in the Methods (i.e.., plant growth, nutrient uptake, etc.). Please list and explain each single parameter/variable used in the study in the Methods.

The statistical methods subsection also need some clarification. For each variable, specify the test used for comparisons. For instance, what about the tests used to compare plant biomass? Why are not mentioned here? And so on...

Results:

Figure 4 is not necessary. Use a Table instead.

Lines 187-201: data in Figure 5 is the same that those in Supplementary Material but just pulled together?  If so, it’s not necessary to show twice. Identify panel A and B in the figure. Specify the test used when you report a p-value (this comment applies for the entire section).

Lines 215-219: how did you compare the treatments? Specify the test used.

Figure 6: how did you measure plant biomass? Explain it in the Methods section. Are there replicates? If so, show SD values in the Figure.

Line 236-238: after revisiting the “objectives” of the paper, I didn’t see that you aimed to investigate the effect of plastic covering on the composition of plant community. Clarify it.

Line 245-250: how did you measure plant nutrient uptake? Explain it in the Methods section. After revisiting the “objectives” of the paper, I didn’t see that you aimed to measure plant uptake in the ditches. Clarify it.

Line 272: in line to comments above, I didn’t see that you aimed compile results from other studies in the literature to be compared to observed results. Please mention that in the study objectives and explain how did you approach it in the Methods section.

Lines 293-300: move this calculation and their rationale to the Method section. Is that another specific objective of the study? Clarify it.

Line 291-293: data interpretation should be move to the Discussion section.

Figure 7: “slope” is the x-axis? Please clarify it.

Figure 8: x-axis range should be 1-12.

Lines 312-315: once again, I didn’t see that you aimed investigate the temporal variation of N and P retention in streams until now. Please include this objective in the Introduction section and the methodology used to approach it in the Methods section. By the way, this objective it is far away from the a priori objective of the study that seems to investigate how watercress and temperature may affect N and P retention in streams. This is so confusing. You need to clearly state the objective(s) of the study.

Discussion:

Lines 317-318: where do you observed an increase of N and P retention? Please, clarify it.

Lines 328-330: these 2 statements needs to be accompanied by some references.

Lines 331-334: Is the first time that you mention a correlation analysis. Please, include this information in the Methods and Results section.

Lines 337-339: I would say that the potential effects of plant diversity on salmonids is out of the scope of this study. You should focus on the potential effects on plant diversity on nutrient removal.

Lines 340-341:  I’m totally agree with you that the use of plastic is not recommended. Therefore, why you used this method? Is there any alternative? On the other hand, I’m wondering if increasing the air temperature along the watercourses using plastic (or other “green” material) is a realistic management practice to enhance nutrient removal in agricultural landscapes... 

Lines 344-359: have you measured PEITC in your study streams? You also refer to the effect of PEITC on macroinvertebrate organisms. Have you also aimed to study the effect of watercress on the in-stream fauna? Please clarify it. If not, this explanation is completely out of the scope.

Lines 363-369: this is an interesting consideration. However, I would say that the effect of watercress on the hydrologic regime is not one of the multiples objectives addressed here. Therefore, you should focus on the effect of watercress on nutrient removal. Alternatively, you may discuss the potential effects of high/low flow on nutrient retention. There are a lot of papers addressing this topic. 

Lines 383-390: there is a lack of references here. Your statements should be supported by other studies. Otherwise is too much speculative.

Finally, I would like to suggest some other related studies that may contribute to enrich this study:

Franklin, P., M. Dunbar, and P. Whitehead. 2008. Flow controls on lowland river macrophytes: A review. Science of the Total Environment.

Gacia, E., S. Bernal, M. Nikolakopoulou, E. Carreras, L. Morgado, M. Ribot, M. Isnard, A. Sorolla, F. Sabater, and E. Martí. 2019. The role of helophyte species on nitrogen and phosphorus retention from wastewater treatment plant effluents. Journal of Environmental Management 252:109585.

Levi, P. S., T. Riis, A. B. Alnøe, M. Peipoch, K. Maetzke, C. Bruus, and A. Baattrup-Pedersen. 2015. Macrophyte complexity controls nutrient uptake in lowland streams. Ecosystems 18:914–931.

Maine, M. A., N. Suñe, H. Hadad, G. Sánchez, and C. Bonetto. 2007. Removal efficiency of a constructed wetland for wastewater treatment according to vegetation dominance. Chemosphere 68:1105–1113.

O’Brien, J. M., J. L. Lessard, D. Plew, S. E. Graham, and A. R. McIntosh. 2014. Aquatic Macrophytes Alter Metabolism and Nutrient Cycling in Lowland Streams. Ecosystems 17:405–417.

Riis, T., A. M. Suren, B. Clausen, and K. A. J. Sand-Jensen. 2008. Vegetation and flow regime in lowland streams. Freshwater Biology 53:1531–1543.

Reviewer 3 Report

The paper is written in an interesting and updated research thematic area. 

You need to improve the bottom-line argument in the connection between citizen science and farmers' actions. As farmers' perception is one of the key aspects of knowledge construction to react to environmental pollution, it is important to discuss that aspect.  Here, it is important to mention the farmers' perception of environmental risk due to pollution. Please read the following paper and cite as necessary 

Withanachchi, S.S.; Kunchulia, I.; Ghambashidze, G.; Al Sidawi, R.; Urushadze, T.; Ploeger, A. Farmers’ Perception of Water Quality and Risks in the Mashavera River Basin, Georgia: Analyzing the Vulnerability of the Social-Ecological System through Community Perceptions. Sustainability 201810, 3062.

Okumah, M., & Yeboah, A. S. (2019). Exploring stakeholders’ perceptions of the quality and governance of water resources in the Wenchi municipality. Journal of Environmental Planning and Management, 1-29.

I recommend making subtitles in the discussion section rather put all arguments in a single section. The argument of citizen science should be here also discussed. 

The conclusion is limited and not enough reflect the paper overal argument.